# Temporal trends of SARS-CoV-2 seroprevalence during the first wave of the COVID-19 epidemic in Kenya

Ifedayo M. O. Adetifa [1,2,7 ✉], Sophie Uyoga [1,7 ✉], John N. Gitonga[1,7], Daisy Mugo[1,7], Mark Otiende[1], James Nyagwange[1], Henry K. Karanja[1], James Tuju[1], Perpetual Wanjiku[1], Rashid Aman[3], Mercy Mwangangi[3], Patrick Amoth[3], Kadondi Kasera[3], Wangari Ng'ang'a[4], Charles Rombo[5], Christine Yegon[5], Khamisi Kithi[5], Elizabeth Odhiambo[5], Thomas Rotich[5], Irene Orgut[5], Sammy Kihara[5], Christian Bottomley [2], Eunice W. Kagucia [1], Katherine E. Gallagher [1,2], Anthony Etyang[1], Shirine Voller[1,2], Teresa Lambe [6], Daniel Wright [6], Edwine Barasa[1], Benjamin Tsofa[1], Philip Bejon[1,6], Lynette I. Ochola-Oyier[1], Ambrose Agweyu[1], J. Anthony G. Scott [1,2,6] & George M. Warimwe [1,6]

Observed SARS-CoV-2 infections and deaths are low in tropical Africa raising questions about the extent of transmission. We measured SARS-CoV-2 IgG by ELISA in 9,922 blood donors across Kenya and adjusted for sampling bias and test performance. By 1st September 2020, 577 COVID-19 deaths were observed nationwide and seroprevalence was 9.1% (95% CI 7.6-10.8%). Seroprevalence in Nairobi was 22.7% (18.0-27.7%). Although most people remained susceptible, SARS-CoV-2 had spread widely in Kenya with apparently low associated mortality.

[1] KEMRI-Wellcome Trust Research Programme, Kilifi, Kenya. [2] Department of Infectious Diseases Epidemiology, London School of Hygiene and Tropical Medicine, Keppel Street, London, United Kingdom. [3] Ministry of Health, Government of Kenya, Nairobi, Kenya. [4] Presidential Policy & Strategy Unit, The Presidency, Government of Kenya, Nairobi, Kenya. [5] Kenya National Blood Transfusion Services, Ministry of Health, Nairobi, Kenya. [6] Nuffield Department of Medicine, Oxford University, Oxford, United Kingdom. [7] These authors contributed equally: Ifedayo M.O. Adetifa, Sophie Uyoga, John N. Gitonga, Daisy Mugo. ✉email: IAdetifa@kemri-wellcome.org; SUyoga@kemri-wellcome.org

Across tropical Africa, numbers of cases and deaths attributable to COVID-19 have been substantially lower than those in Europe and the Americas[1]. This could imply reduced transmission, reduced clinical severity or epidemiological under-ascertainment. The first COVID-19 case in Kenya was identified on 12th March 2020. Subsequently, there have been two discrete waves of PCR-detected cases in May–August and October–January separated by a brief nadir in September 2020. At the end of 2020, the government had recorded 96,595 cases and 1792 deaths attributable to SARS-CoV-2[2]. By May 30 2020 when COVID-19 related deaths reached 71, the national anti-SARS-CoV-2 antibody prevalence, estimated in blood donors, was 4.3% (95% confidence interval (CI) 2.9–5.8%)[3]. Transmission was obviously more widespread than would have been anticipated by reported cases and deaths. In this further study, we examine the dynamics of SARS-CoV-2 seroprevalence among Kenyan blood donors throughout the course of the first epidemic wave.

From 30th April to 30th September 2020, 10,258 samples from blood donors aged 16–64 years were processed at six Kenya National Blood Transfusion Service (KNBTS) regional blood transfusion centres, which serve a countrywide network of satellites and hospitals. We excluded duplicate samples, those from age-ineligible donors and those with missing data, leaving 9922 samples (Supplementary Fig. 1).

The blood donor samples were broadly representative of the Kenyan adult population[4] on region of residence and age, although adults aged 55–64 years were under-represented (2.0% vs 7.3%, Supplementary Table 1) and adults aged 25–34 years were over-represented (39.3% vs 27.3%). Males were also over-represented (80.8%).

We tested samples for anti-SARS-CoV-2 IgG antibodies using a previously described ELISA for whole length spike antigen[5]. Assay sensitivity, estimated in sera from 174 PCR positive Kenyan adults and a panel of sera from the UK National Institute of Biological Standards and Control (NIBSC) was 92.7% (95% CI 87.9–96.1%); specificity, estimated in 910 serum samples from Kilifi drawn in 2018 was 99.0% (95% CI 98.1–99.5%)[3]. Assays on a subset of test samples were repeated at least once on separate days and reproducibility confirmed. Positive and negative control samples are routinely included in all runs and the results from these were reproducible.

## Results and discussion

Of the 9922 samples with complete data 3098 had been reported previously[3]. In total, 928 were positive for anti-SARS-CoV-2 IgG; crude seroprevalence was 9.4% (95% CI, 8.8–9.9%) with little variation by age or sex (Table 1). We used Bayesian Multi-level Regression with Post-stratification (MRP) to adjust for test sensitivity (93%) and specificity (99%)[6], smooth trends over time, and account for the differences in age, sex and residence characteristics of the test sample and the Kenyan population[7].

There was marked variation in seroprevalence over time and place with a generally increasing trend over time. Figures 1 and 2 respectively illustrate the cumulative confirmed COVID-19 cases in Kenya during the study period and the crude prevalence and Bayesian model estimates in 10 consecutive periods of ~2 weeks each. In Nairobi, Mombasa and the Coastal Region outside Mombasa, there was a steep rise in seroprevalence across the study period. We divided the observations equally into three consecutive periods (Table 2). In period 1 (30 April–19 June) the adjusted seroprevalence of SARS-CoV-2 was 5.2% (95% CI 3.7–6.7%); in period 2 (20 June–19 August) it had risen to 9.1% (95% CI 7.2–11.3%); and in period 3 (20 August–30 September) it was maintained at 9.1% (95% CI 7.6–10.8%).

The results illustrate a heterogeneous pattern of transmission across Kenya and suggest that the seroprevalence first began to rise in Mombasa in May and reached a maximum in July; in Nairobi it increased steadily from June onwards; in the Coastal area seroprevalence began to rise in July and turned up sharply in August and September. Unlike Nairobi and Mombasa this area is mostly rural. Other parts of the country showed less of a temporal trend. These field observations accord closely with epidemic modelling of SARS-CoV-2 across Kenya which integrated early PCR and serological data with mobility trends to describe the transmission pattern nationally[8].

Although we used a highly specific and validated assay[3,9], and adjusted for biases inherent in the ELISA test performance, we did not control for antibody waning. Given evidence at both individual[10] and population[11] level that anti-Spike antibodies may decline after an initial immune response, cross-sectional data are likely to underestimate cumulative incidence with increasing error as the epidemic wave declines. Some investigators have adjusted for this effect through modelling[12] but as we do not have a clear description of the waning function for these antibodies in our setting, we have not made such an adjustment. We are exploring the application of mixture modelling to account for this challenge[13]. Therefore, the seroprevalence estimates reported here are likely to underestimate cumulative incidence in Kenya.

The study also relies on convenience sampling of asymptomatic blood transfusion donors which is not representative of the adult population at large and may underestimate seroprevalence because those with a recent history of illness are excluded. Although blood donors are predominantly male in Kenya, we had ~2000 female donor samples and stratified all analyses by sex to ensure that any potential confounding was appropriately adjusted for. We have adjusted for demographic and geographic disparities in our sample set, but we are unable to evaluate whether the behaviour of blood donors increases or reduces their risk of infection by SARS-CoV-2. The exclusion of donors with history of illness in the past 6 months may also contribute to selection bias. A random population sample would overcome these problems, but such studies were difficult to undertake during movement restrictions[8]. Recruiting household contacts of blood donors was considered but this was beyond the remit of the KNBTS and the movement and other restrictions also made this impractical. The selection bias in KNBTS samples is unlikely to change substantially over time and therefore this survey and the continued surveillance of blood donors will provide valid estimation of trends, which inform the public health management of the epidemic.

The results are also consistent with other surveys in Kenya which have illustrated both high seroprevalence in focal populations and marked geographic variation. For example, seroprevalence was 50% among women attending ante-natal care (ANC) in August 2020 in Nairobi but 1.3%, 1.5% and 11.0% among women attending ANC in Kilifi (Coast) in September, October and November, respectively[14]. Seroprevalence among truck drivers at two sites (in Coast and Western) was 42% in October 2020[15] and seroprevalence among health care workers in between August and November 2020 was 43%, 12% and 11% in Nairobi, Busia (Western) and Kilifi (Coast), respectively[16].

By 1st September 2020, the first epidemic wave of SARS-CoV-2 in Kenya had declined with a cumulative mortality of 767 COVID-19 deaths and 34,471 cases.[2] Our large national blood donor serosurvey illustrates that, at the same point, 1 in 10 donors had antibody evidence of infection with SARS-CoV-2; this rises to 1 in 5 in the two major cities in Kenya. The first epidemic

**Table 1 Crude, age/sex standardised and Bayesian-weighted test-adjusted SARS-CoV-2 anti-spike protein IgG seroprevalence across the whole study duration.**

| | Kenya population | All samples (%) | Sero-positive | Crude seroprevalence | | Bayesian weighted, test-adjusted seroprevalence[a] | |
|---|---|---|---|---|---|---|---|
| | | | | % | 95% CI | % | 95% CI |
| **Age** | | | | | | | |
| 15–24 years | 9,733,174 | 2763 (27.8) | 241 | 8.7 | 7.7–9.8 | 7.5 | 6.2–8.8 |
| 25–34 years | 7,424,967 | 3902 (39.3) | 379 | 9.7 | 8.8–10.7 | 8.5 | 7.2–9.8 |
| 35–44 years | 4,909,191 | 2261 (22.8) | 224 | 9.9 | 8.7–11.2 | 8.3 | 6.9–9.8 |
| 45–54 years | 3,094,771 | 794 (8.0) | 66 | 8.3 | 6.5–10.5 | 7.3 | 5.5–8.9 |
| 55–64 years | 1,988,062 | 202 (2.1) | 18 | 8.9 | 5.4–13.7 | 7.2 | 5.2–9.1 |
| **Sex** | | | | | | | |
| Male | 13,388,243 | 8019 (80.8) | 762 | 9.5 | 8.9–10.2 | 8.4 | 7.2–9.5 |
| Female | 13,761,922 | 1903 (19.2) | 166 | 8.7 | 7.5–10.1 | 7.4 | 5.9–8.9 |
| **Region** | | | | | | | |
| Central | 3,452,213 | 606 (6.1) | 38 | 6.3 | 4.5–8.5 | 5.8 | 3.7–8.0 |
| Coast | 1,671,097 | 1680 (16.9) | 137 | 8.2 | 6.9–9.6 | 7.2 | 5.6–8.9 |
| Eastern/N. Eastern | 5,176,080 | 1482 (14.9) | 108 | 7.3 | 6.0–8.7 | 6.5 | 4.9–8.2 |
| Mombasa | 792,072 | 1654 (16.7) | 239 | 14.4 | 12.8–16.2 | 13.8 | 11.7–16 |
| Nairobi | 3,002,314 | 607 (6.1) | 107 | 17.6 | 14.7–20.9 | 16.7 | 13.4–20.2 |
| Nyanza | 3,363,813 | 1433 (14.4) | 131 | 9.1 | 7.7–10.8 | 8.3 | 6.6–10.2 |
| Rift Valley | 7,035,581 | 2138 (21.6) | 145 | 6.8 | 5.8–7.9 | 5.9 | 4.5–7.4 |
| Western | 2,656,995 | 322 (3.3) | 23 | 7.1 | 4.6–10.5 | 6.6 | 3.9–9.7 |
| National | 27,150,165 | 9922 (100) | 928 | 9.4 | 8.8–9.9 | 7.9 | 6.7–9.0 |

[a]Bayesian Multi-level Regression with Post-stratification (MRP) accounts for differences in the age and sex distribution of blood donors and regional differences in the numbers of samples collected over time. The model also adjusts for sensitivity (93%) and specificity (99%) of the ELISA.

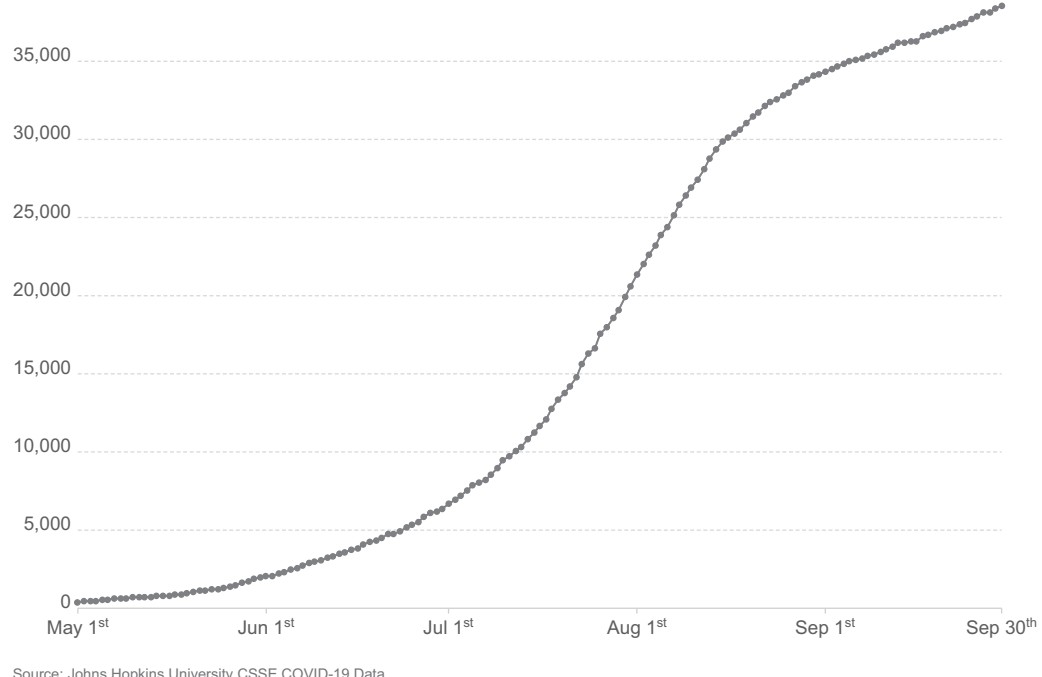

Source: Johns Hopkins University CSSE COVID-19 Data

**Fig. 1 Cumulative confirmed COVID-19 cases in Kenya from 1st May - 20th September 2020.**

wave rose and fell against a background of constant movement restrictions. The seroprevalence estimates suggest that population immunity alone was inadequate to explain this fall and majority of the population remained susceptible. Nonetheless, they also show that the virus was widely transmitted during the first epidemic wave even though numbers of cases and deaths attributable to SARS-CoV-2 in Kenya were very low by comparison with similar settings in Europe and the Americas at similar seroprevalence[17,18]. This pattern of widespread SARS-CoV-2 transmission and higher cumulative exposure in general[19–22] and targeted populations (including blood donors)[23–26] compared to disproportionately lower COVID-19 case numbers and deaths has also been seen across epidemic waves in other parts of Africa. This disparity may be attributable to constraints on morbidity/

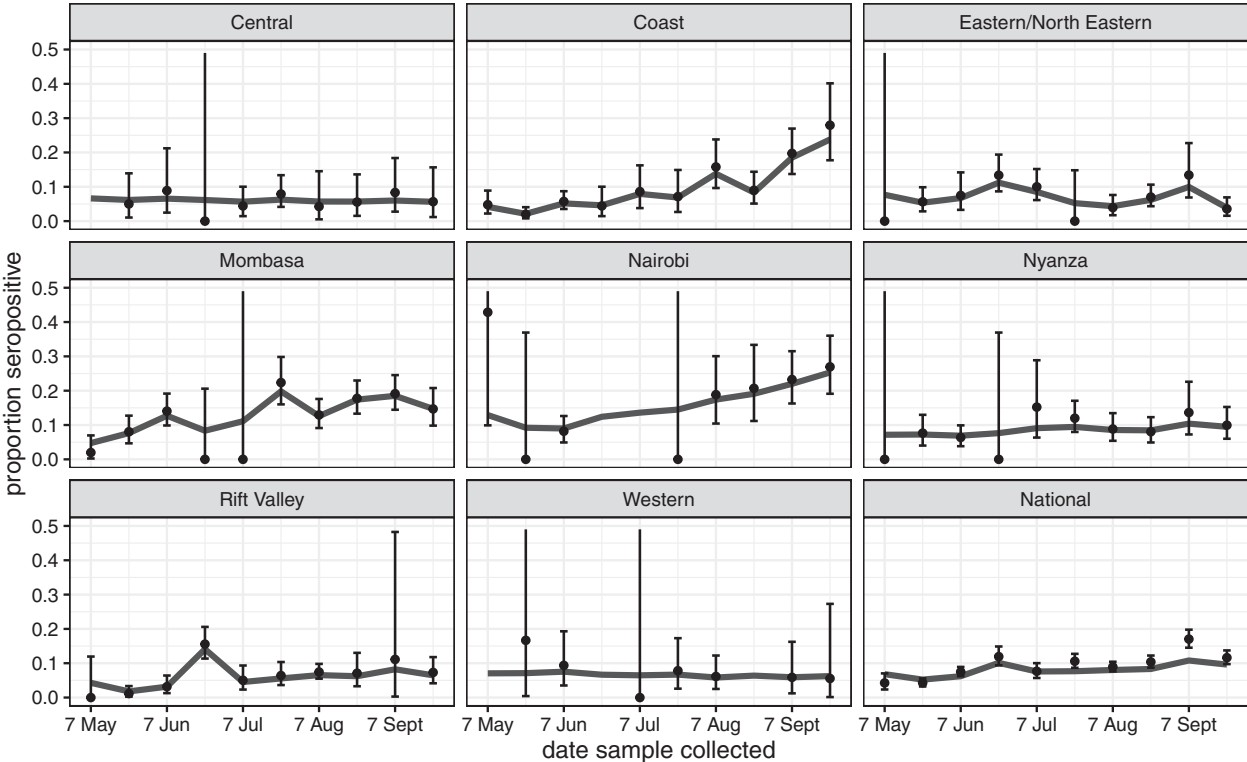

**Fig. 2 Seroprevalence positivity across the study period by region.** The figure shows unadjusted estimates (black dots) and Bayesian model estimates (grey line) of seroprevalence in 8 regions of Kenya and overall, by date of sample collection in 10 periods of ~2 weeks each during 2020 (n = 9992). With the exception of the first period in Rift Valley (7 May) and 6th period in Eastern/North Eastern (21 July), all data estimates of zero prevalence are based on small sample sets (<20 samples). Error bars are 95% Confidence Intervals.

**Table 2 Bayesian-weighted test-adjusted SARS-CoV-2 anti-spike protein IgG seroprevalence across three study periods.**

| | Period 1 (30 Apr–19 Jun) | | Period 2 (20 Jun–19 Aug) | | Period 3 (20 Aug–30 Sep) | |
|---|---|---|---|---|---|---|
| | Prevalence | 95% CIª | Prevalence | 95% CI | Prevalence | 95% CI |
| Age | | | | | | |
| 15–24 years | 5.2 | 3.4–7.1 | 8.3 | 5.9–10.8 | 8.7 | 6.9–10.6 |
| 25–34 years | 5.2 | 3.6–7.0 | 9.3 | 7.1–12.0 | 10.2 | 8.5–12.4 |
| 35–44 years | 6.1 | 4.1–8.6 | 10.5 | 7.7–13.8 | 8.7 | 6.8–10.7 |
| 45–54 years | 4.1 | 1.6–6.5 | 9.2 | 6.3–12.8 | 8.6 | 6.2–11.0 |
| 55–64 years | 4.1 | 1.1–6.9 | 8.7 | 4.6–12.9 | 8.8 | 6.3–12.9 |
| Sex | | | | | | |
| Male | 6.1 | 4.6–7.6 | 8.4 | 6.7–10.2 | 10.1 | 8.6–11.7 |
| Female | 4.3 | 2.3–6.5 | 9.7 | 6.9–13.1 | 8.2 | 6.1–10.5 |
| Region | | | | | | |
| Central | 5.3 | 2.5–9.0 | 6.9 | 3.9–10.4 | 5.9 | 3.0–9.5 |
| Coast | 3.3 | 1.9–4.8 | 11.3 | 7.4–16.0 | 14.1 | 10.6–18.1 |
| Eastern/N. Eastern | 5.4 | 3.2–8.2 | 9.6 | 6.7–12.9 | 5.2 | 3.4–7.3 |
| Mombasa | 7.4 | 5.0–10.1 | 17.2 | 12.5–23.0 | 15.9 | 13.1–19.1 |
| Nairobi | 6.7 | 3.9–10.5 | 10.1 | 3.7–20.1 | 22.7 | 18–27.7 |
| Nyanza | 5.3 | 3.2–7.7 | 11.3 | 8.0–15.2 | 8.5 | 6.1–11.3 |
| Rift Valley | 3.9 | 2.3–5.8 | 7.3 | 5.3–9.6 | 7.3 | 4.9–10.0 |
| Western | 6.3 | 2.9–11.6 | 7.7 | 3.9–12.2 | 6.0 | 2.4–11.1 |
| National | 5.2 | 3.7–6.7 | 9.1 | 7.2–11.3 | 9.1 | 7.6–10.8 |

ª95% credible interval.

mortality surveillance and poor availability of PCR testing and, to some degree, to the different age structures of African and European populations.

These and our other estimates of cumulative incidence of SARS-CoV-2 support the need for the SARS-CoV-2 vaccination in Kenya since a significant proportion of the population remains susceptible to infection and COVID-19 disease. In addition, vaccination will interrupt transmission, prevent the development of variants, and ultimately correct the social and economic disruption caused by this pandemic.

## Methods

**Human samples**. The Kenya National Blood Transfusion Service (KNBTS) coordinates and screens blood transfusion donor units at 6 regional centres at Eldoret, Embu, Kisumu, Mombasa, Nairobi and Nakuru, though the units are collected across the whole country and each Regional Centre serves between 5 and 10 of Kenya's 47 Counties. KNBTS guidelines define eligible blood donors as individuals aged 16–65 years, weighing ≥50 kg, with haemoglobin of 12·5 g/dl, a normal blood pressure (systolic 120–129 mmHg and diastolic BP of 80–89 mmHg), a pulse rate of 60–100 beats per minute and without any history of illness in the past 6 months[27]. KNBTS generally relies on voluntary non-remunerated blood donors (VNRD) recruited at public blood drives typically located in high schools, colleges and universities. Since September 2019, because of reduced funding, KNBTS has depended increasingly on family replacement donors (FRD) who provide units of blood in compensation for those received by sick relatives. We obtained anonymized residual samples from consecutive donor units submitted to the 6 regional centres for transfusion compatibility-testing and infection screening, as previously described[3].

**Laboratory analyses**. Enzyme linked Immunosorbent Assay (ELISA) IgG antibodies to the SARS-CoV-2 spike protein were measured using a previously described ELISA at the KEMRI-Wellcome Trust Research Programme in Kilifi, Kenya. Following a validation exercise and estimate of sensitivity and specificity, results were expressed as the ratio of test OD to the OD of the plate negative control; samples with OD ratios greater than two were considered positive for SARS-CoV-2 IgG[3,5]. In a WHO-sponsored multi-laboratory study of SARS-CoV-2 antibody assays, results from Kilifi were consistent with the majority of the test laboratories[9].

**Statistical analysis**. We estimated crude prevalence based on the proportion of samples with OD ratio > 2. We also used Bayesian Multi-level Regression with Post-stratification (MRP)[7] to account for differences in the age and sex distribution of blood donors and regional differences in the numbers of samples collected over time. Data on donor residence were specified at County level. For the purposes of analysis and presentation we collapsed the 47 counties into 8 regions based on the previous administrative provinces of Kenya; as data from two regions (Eastern and North Eastern) was relatively sparse we collapsed these to one stratum. The model was also used to adjust for sensitivity (93%) and specificity (99%) of the chosen cut-off value as previously developed[6]. Regional and national estimates were produced by combining model predictions with weights from the 2019 Kenyan census[4]. Two versions of the model were fitted. In the first (Model A), the model included age, sex and region as covariates and was fitted separately to data in three periods (30 Apr–19 Jun, 20 Jun–19 Aug, 20 Aug–30 Sept). In the second (Model B), the model also included a period effect and was fitted to the samples as a whole. A mathematical description of the models and Rstan code[28] is provided in the statistical appendix.

**Ethical approval**. This study was approved by the Scientific and Ethics Review Unit (SERU) of the Kenya Medical Research Institute (Protocol SSC 3426). Blood donors gave individual written consent for the use of their samples for research.

**Reporting summary**. Further information on research design is available in the Nature Research Reporting Summary linked to this article.

## Data availability

The raw data shown in the manuscript are subject to controlled access because they are the subject of ongoing work and will be made available on request to the corresponding author and approval by the Data Governance Committee at the KEMRI-Wellcome Trust Research Programme. De-identified data has been published on the Harvard dataverse server https://doi.org/10.7910/DVN/FQUNVD.

## Code availability

Code related to the Bayesian Multi-level Regression with Post-stratification can be found in the Supplementary note 1: statistical appendix accompanying this article.

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

## Acknowledgements

This project was funded by the Wellcome Trust (grant numbers 220991/Z/20/Z, 203077/Z/16/Z), the Bill and Melinda Gates Foundation (INV-017547) and by the Foreign Commonwealth and Development Office (FCDO) through the East Africa Research Fund (EARF/ITT/039). S.U. is funded by DELTAS Africa Initiative [DEL-15-003]. L.I.O.-O. is

funded by a Wellcome Trust Intermediate Fellowship (107568/Z/15/Z), A.A. is funded by a DFID/MRC/NIHR/Wellcome Trust Joint Global Health Trials Award (MR/R006083/1), J.A.G.S. is funded by a Wellcome Trust Senior Research Fellowship (214320) and the NIHR Health Protection Research Unit in Immunisation, I.A. is funded by the United Kingdom's Medical Research Council and Department For International Development through a African Research Leader Fellowship (MR/S005293/1) and by the NIHR-MPRU at UCL (grant 2268427 LSHTM). G.M.W. is supported by a fellowship from the Oak Foundation. For the purpose of Open Access, the author has applied a CC-BY public copyright licence to any author accepted manuscript version arising from this submission. This paper has been published with the permission of the Director, Kenya Medical Research Institute. We thank all of the blood donors for their contribution to the research.

## Author contributions

I.M.O.A., S.U., A.A., J.A.G.S. and G.W. designed the study and did the literature review. E.W., K.G., A.E., S.V., R.A., M.M., P.A., K.K. and W.N. contributed to the study design, protocol development and implementation. S.U., P.W., C.R., C.Y., K.K.I., E.O., T.R., I.O. and S.K. helped collect the samples and demographic data. T.L., D.W., H.K.K., J.N., J.T., L.I.O.-O. and G.W. developed and validated the ELISA assay. J.N.G., D.M., H.K.K., J.N., J.T. and L.I.O.-O. were responsible for sample preparation and laboratory analyses. E.B., B.T. and P.B. provided coordination and contributed to supervision. M.O., C.B. and J.A.G.S. analysed data and all authors interpreted data. S.U., I.M.O.A., J.A.G.S., A.A. and G.W. wrote the first draft of the report and all authors contributed to editing the final version. A.A., J.A.G.S. and G.W. contributed equally as senior authors.

## Competing interests

R.A., M.M., K.K. and P.A. are from the Ministry of Health, Government of Kenya. All other authors declare no competing interests.
