## [Peer Review File · Nature Communications]

REVIEWER COMMENTS

Reviewer #1 (Remarks to the Author):

The authors report a seroprevalence of 9.1% in Kenya, and 22.7% in Nairobi by 1st September 2020. They claim that SARS-CoV-2 had spread widely in Kenya but with apparently low associated mortality.

Overall, the study was well-conducted with methodology well-stated and all the data presented within the submission. The results are not novel from the Kenyan perspective, as similar studies from this population have been published by the authors and others, however the data provides further evidence of the unusual trajectory of the COVID-19 pandemic in Africa.

The manuscript would benefit with revision in following areas:

Major:

1. The paper would benefit with a discussion on the how their results map within the wider African context. There has been a couple of studies published from other sites in Africa that are not discussed in this manuscript. It would be helpful for the authors to contextualise their results beyond Kenya, into the rest of Africa, as this could help development of locally-relevant public health strategies.
2. The authors should include in their discussion the implication of these results on vaccination programmes in Kenya, and in similar populations elsewhere.

Minor:

1. The authors should include confidence intervals in the graphs shown in Figure 1.
2. The authors should comment on whether using a smaller sample size by random sampling of the blood donor sera could have generated similar results.

Reviewer #2 (Remarks to the Author):

Adetifa et al. examined the dynamics of SARS-CoV-2 seroprevalence among Kenyan blood donors throughout the course of the first epidemic wave. The manuscript provides interesting data and can contribute to help the community further understand the epidemiology of the pandemic. I believe the manuscript can be improved and thus provide some suggestions (I wish the MS had pages and lines).

Major comments:

Readers should be able to interpret the dynamics of the seroprevalence in the light of the dynamics of the cases and cumulative cases, figures and discussion should superpose the two information.

Data are derived from blood donors. More discussion on the possible direction and estimation of selection bias should be provided. Authors make some suggestions, but one criteria of exclusion of donors was "any history of illness in the past 6 months", authors should clearly discuss how this could influence the estimations (as most SARS positive would be sick and consider themselves as ill and not eligible)?

So all participants were asymptomatic?

The authors stress the challenge of collecting population-based data during this period, why not inviting all members of the donors' household to join the study?

The gender disequilibrium between blood transfusion samples and National Census is very concerning as well as the fact that very small regions (census speaking) are largely represented in the transfusion samples. Authors should further comment and consider further adjustment to reassure the readers. For example, did the authors post-stratified the modelled results accounting for the age and sex distribution to generate population-representative seroprevalence estimates?

More information is now available on the waning function of antibodies, adjusting some of the results would be outstanding.

Minor comments:

Main text first paragraph:

References are needed in the first paragraph, not clear where the data and comments are coming from.

More information on moving restriction policy in Kenya during the study period is needed

WHO recommends monitoring changes of seroprevalence over time to plan an adequate public health response, what was the response in Kenya (physical distancing and confinement measures)?

Main text second paragraph:

The authors should help the readers disentangle the first to any other waves that happened in Kenya. My understanding is that a second wave happened, more information is needed.

Even though it is mentioned in the supplement I suggest to report the age of eligibility for donors (16-65)

Not clear whether the samples were consecutive cases.

Main text third paragraph:

Can the authors provide a reference on the assay performances they reported?

Main text 10th paragraph:

Here readers should understand when the second wave started in Kenya, and should have a better understanding of the dynamic of cases during the periods

Additional comments :

What is planned to assess the seroprevalence among young adults and elderly ?

Is it correct that there was no IRB approval needed ?

Did the authors use a confirmatory test (recombinant immunofluorescence assay) potentially indeterminate individuals ?

Important to stress that model did not account for random effect for household

Did the authors run iterations and assessed convergence in their models?

Why didn't the authors calculate and provide the relative risk (RR) of being seropositive?

Did the authors consider using their age-specific seroprevalences estimate the infection fatality risk?

Figure 1:

Estimate should include CI (this would help interpret the distance between raw and modeled estimations)

Authors should comment on the major distance between raw and modeled estimates (clearly illustrated in Figure 1).

Suppl Table 2:
the voluntary and family subtype must be reported in the methods

Table 1a:
Suggest to add % to the All samples column

Reviewer #3 (Remarks to the Author):

This is a well written report on a national surveillance programme for SARS-CoV-2 antibodies using data from blood donors. The overall finding is of a higher prevalence than might be expected from the apparent scale of the epidemic based on routine surveillance of cases and deaths in Kenya, but the significance and interpretation of this finding is not explored.

The manuscript would be strengthened by situating it in the wider literature on the relationship between cases, deaths and prevalence, particularly in other low income countries. The comparison drawn at the end is to high income countries, where it is suggested that with a similar antibody prevalence there has been a far higher mortality. It would be important to contextualise this in terms of (a) testing availability (b) case, hospitalisation and mortality surveillance systems, and (c) possible explanations such as the different age structures of the populations.

RESPONSE TO REVIEWERS' COMMENTS- NCOMMS-21-06923-T

Reviewer #1 (Remarks to the Author):

The authors report a seroprevalence of 9.1% in Kenya, and 22.7% in Nairobi by 1st September 2020. They claim that SARS-CoV-2 had spread widely in Kenya but with apparently low associated mortality.

Overall, the study was well-conducted with methodology well-stated and all the data presented within the submission. The results are not novel from the Kenyan perspective, as similar studies from this population have been published by the authors and others, however the data provides further evidence of the unusual trajectory of the COVID-19 pandemic in Africa.

Thank you for these kind comments

The manuscript would benefit with revision in following areas:

Major:

1. The paper would benefit with a discussion on the how their results map within the wider African context. There has been a couple of studies published from other sites in Africa that are not discussed in this manuscript. It would be helpful for the authors to contextualise their results beyond Kenya, into the rest of Africa, as this could help development of locally-relevant public health strategies.

Page 6 Lines 84-85 and page 7 lines 87-91. We have now indicated that studies from other African settings replicate our results i.e., extensive SARS-CoV-2 transmission in the population in far excess of reported COVID-19 case numbers and deaths in general and targeted populations.

2. The authors should include in their discussion the implication of these results on vaccination programmes in Kenya, and in similar populations elsewhere.

We have added text as suggested from Page 7 lines 92-96.

Minor:

1. The authors should include confidence intervals in the graphs shown in Figure 1.

These have now been included.

2. The authors should comment on whether using a smaller sample size by random sampling of the blood donor sera could have generated similar results.

Seroprevalence estimates from a probability-based population sample is the preference as an epidemiological tracker for this pandemic but the paucity of such data highlights the challenge of setting up these surveys in Kenya and other African countries especially at the onset of the pandemic. Blood donors represent a convenience sample whose results

also have utility for validation of community-level seroprevalence estimates. While a random sample of blood donors could have been considered, it would not have been possible to conduct such a study in Kenya since residual blood samples from donors are not routinely banked to provide a sampling frame, and we processed all the samples that were made available to us.

We have not made any changes to the manuscript on this specific point but would be happy to follow further advice if this justification is felt necessary to include.

Reviewer #2 (Remarks to the Author):

Adetifa et al. examined the dynamics of SARS-CoV-2 seroprevalence among Kenyan blood donors throughout the course of the first epidemic wave. The manuscript provides interesting data and can contribute to help the community further understand the epidemiology of the pandemic. I believe the manuscript can be improved and thus provide some suggestions (I wish the MS had pages and lines).

Thank you for these comments. We have now included line and page numbers

Major comments:

Readers should be able to interpret the dynamics of the seroprevalence in the light of the dynamics of the cases and cumulative cases, figures and discussion should superpose the two information.

Page 6 line 79 and figure 1A. We agree and our inclusion of COVID19 case numbers to complement the earlier provided numbers for cumulative deaths now allows for a better understanding of the dynamics of these epidemiological trackers at the same point in time.

Data are derived from blood donors. More discussion on the possible direction and estimation of selection bias should be provided. Authors make some suggestions, but one criteria of exclusion of donors was “any history of illness in the past 6 months”, authors should clearly discuss how this could influence the estimations (as most SARS positive would be sick and consider themselves as ill and not eligible)?

On Page 5 lines 59 to 64, we discuss selection bias, our efforts to address this and conclude our description of trends in this manuscript is valid since this bias is expected to be constant in our blood donor population over the duration of our data collection. It is difficult for us to assess how the “history of illness in the past 6 months” criteria is implemented in practice, given that mild febrile episodes are common and often not recalled. Furthermore, most SARS-CoV-2 infection in Kenya appears to be asymptomatic. Nevertheless, it is possible that this exclusion criteria leads us to underestimate the population seroprevalence and we have also added this to the text on selection bias on page 5 lines 62 to 64.

So all participants were asymptomatic?

Yes, they were at the time of blood donation.

The authors stress the challenge of collecting population-based data during this period, why not inviting all members of the donors' household to join the study?

The movement and other restrictions in place also prevented this from happening, and the blood donation service would not have been in a position to implement this sampling. Our priority for when restrictions can be overcome would be the gold-standard of a probability-based population sample.

The gender disequilibrium between blood transfusion samples and National Census is very concerning as well as the fact that very small regions (census speaking) are largely represented in the transfusion samples.

It is true that blood donors are predominantly male in Kenya. However, we assayed ~2000 female donor samples and stratified all analyses by sex to ensure that any potential confounding was appropriately adjusted for. We interpret the regional distribution differently. The samples available are well distributed across regions for example, Rift Valley, Nyanza and Coast (including Mombasa), Central, Western and Nairobi (see table 1A).

Authors should further comment and consider further adjustment to reassure the readers. For example, did the authors post-stratified the modelled results accounting for the age and sex distribution to generate population-representative seroprevalence estimates?

In the methods on page 4 lines 29-32, we describe the adjustments made to the data including accounting for the gender disequilibrium. "We used Bayesian Multi-level Regression with Post-stratification (MRP) to adjust for test sensitivity (93%) and specificity (99%)⁶, smooth trends over time, and account for the differences in age, sex and residence characteristics of the test sample and the Kenyan population⁷."

More information is now available on the waning function of antibodies, adjusting some of the results would be outstanding.

The waning function of antibodies in our study or similar setting is still unknown and waning adjustments in other settings have proven controversial (page 5 line 56, reference 12). Rather than adjust our results for this through modelling, we have opted to present them as they are. We acknowledge our seroprevalence results may be an underestimate of cumulative incidence in Kenya (page 5 lines 52-58). We have also used our data to explore the application of mixture modelling approaches to account for this challenge and describe this in a manuscript to be submitted shortly.

Minor comments:

Main text first paragraph:

References are needed in the first paragraph, not clear where the data and comments are coming from.

We have now provided additional references on page 1 lines 3 and 8

More information on moving restriction policy in Kenya during the study period is needed

This has now been addressed through reference 8 on page 6 line 64.

WHO recommends monitoring changes of seroprevalence over time to plan an adequate public health response, what was the response in Kenya (physical distancing and confinement measures)?

Physical distancing and restrictions on activity have been used in Kenya, and these have been done taking into account the rates of severe disease and death as well as seroprevalence. More details on the measures are as described in reference 8 on page 6 line 64.

Main text second paragraph:

The authors should help the readers disentangle the first to any other waves that happened in Kenya. My understanding is that a second wave happened, more information is needed.

This manuscript under review is confined to the period of the first wave. Our continued serosurveillance of blood donors and other populations will provide updates on the second wave.

Even though it is mentioned in the supplement I suggest to report the age of eligibility for donors (16-65)

This information has now been provided on page 3 line 13

Not clear whether the samples were consecutive cases.

This information can be found on page 13 lines 13-15

Main text third paragraph:

Can the authors provide a reference on the assay performances they reported?

This is reference 3 as shown on page 4 line 26.

Main text 10th paragraph:

Here readers should understand when the second wave started in Kenya, and should have a better understanding of the dynamic of cases during the periods

We agree this is useful information. It is beyond the subject of the manuscript under review which is about the first epidemic wave. We will be able to share data on the second and other waves as we continue our surveillance in blood donors and other populations, but these data are not yet available.

Additional comments :

What is planned to assess the seroprevalence among young adults and elderly?

There are now covered in other surveys by us and other research groups.

Is it correct that there was no IRB approval needed?

Page 14 lines 39-41 shows IRB approval and written informed consent were obtained

Did the authors used a confirmatory test (recombinant immunofluorescence assay) potentially indeterminate individuals?

We did not conduct any further confirmatory testing and there were no potentially indeterminate individuals with our assay.

Important to stress that model did not account for random effect for household

We did not sample by household and did not conduct any random effects regression analyses. As described in the methods on Page 14 lines 24-38, we used Bayesian Multi-level Regression with Post-Stratification to account for differences in the age and sex distribution of blood donors and differences in the numbers of samples collected over time by region/location.

Did the authors ran iterations and assessed convergence in their models?

We simulated 10,000 iterations of 3 chains with a burn in of 1,000 iterations. Convergence was assessed through the R-hat statistic and by visual inspection of the chains.

Why didn't the authors calculate and provide the relative risk (RR) of being seropositive?

Our main objective for this paper was to describe the dynamics of SARS-CoV-2 seroprevalence among blood donors in Kenya during the course of the first epidemic wave.

Did the authors consider using their age-specific seroprevalences estimate the infection fatality risk?

This is covered in another paper i.e., reference 8 on page 5 line 64

Figure 1:

Estimate should include CI (this would help interpret the distance between raw and modeled estimations)

Confidence intervals are now included in this figure

Authors should comment on the major distance between raw and modeled estimates (clearly illustrated in Figure 1).

As the reviewer seems to be suggesting, the discrepancy between modelled and observed estimates is mainly a function of sample size: the discrepancy is greatest for periods where few samples were available. This is illustrated more clearly now through the inclusion of confidence intervals in Figure 1.

Suppl Table 2:
the voluntary and family subtype must be reported in the methods

This was reported on page 13 lines 9-13

Table 1a:
Suggest to add % to the All samples column

This has been done

Reviewer #3 (Remarks to the Author):

This is a well written report on a national surveillance programme for SARS-CoV-2 antibodies using data from blood donors. The overall finding is of a higher prevalence than might be expected from the apparent scale of the epidemic based on routine surveillance of cases and deaths in Kenya, but the significance and interpretation of this finding is not explored.

The manuscript would be strengthened by situating it in the wider literature on the relationship between cases, deaths and prevalence, particularly in other low income countries. The comparison drawn at the end is to high income countries, where it is suggested that with a similar antibody prevalence there has been a far higher mortality. It would be important to contextualise this in terms of (a) testing availability (b) case, hospitalisation and mortality surveillance systems, and (c) possible explanations such as the different age structures of the populations.

Thank you for your kind comments and suggestions which we have now addressed on Page 6 lines 84 to page 7 lines 87-91

REVIEWERS' COMMENTS

Reviewer #2 (Remarks to the Author):

Congratulations on the revision. You have addressed my concerns directly in the rebuttal without mentioning it in your revised MS. I am concerned that other readers will leave your work with the same questions I had but without the elements I read from your rebuttal letter. If word count allows it, I suggest you discuss some of these points in you revised manuscript.

RESPONSE TO REVIEWERS' COMMENTS- NCOMMS-21-06923A

Reviewer #2 (Remarks to the Author):

Congratulations on the revision. You have addressed my concerns directly in the rebuttal without mentioning it in your revised MS. I am concerned that other readers will leave your work with the same questions I had but without the elements I read from your rebuttal letter. If word count allows it, I suggest you discuss some of these points in your revised manuscript.

Thank you very much for your feedback. We have tried to incorporate some of our responses in the rebuttal letter into the manuscript as follows.

1. Readers should be able to interpret the dynamics of the seroprevalence in the light of the dynamics of the cases and cumulative cases, figures and discussion should superpose the two information.

Page 7 line 86 and figure 1A. We agree and our inclusion of COVID19 case numbers to complement the earlier provided numbers for cumulative deaths now allows for a better understanding of the dynamics of these epidemiological trackers at the same point in time.

2. Data are derived from blood donors. More discussion on the possible direction and estimation of selection bias should be provided. Authors make some suggestions, but one criteria of exclusion of donors was "any history of illness in the past 6 months", authors should clearly discuss how this could influence the estimations (as most SARS positive would be sick and consider themselves as ill and not eligible)?

On page 6 lines 62-76, we discuss selection bias, our efforts to address this and conclude our description of trends in this manuscript is valid since this bias is expected to be constant in our blood donor population over the duration of our data collection. It is difficult for us to assess how the "history of illness in the past 6 months" criterion is implemented in practice, given that mild febrile episodes are common and often not recalled. Furthermore, most SARS-CoV-2 infection in Kenya appears to be asymptomatic. Nevertheless, it is possible that this exclusion criteria leads us to underestimate the population seroprevalence and we have also added this to the text on selection bias on page 6 lines 69-70.

3. So, all participants were asymptomatic?

Yes, they were at the time of blood donation. This is mentioned on page 6 line 62.

4. The authors stress the challenge of collecting population-based data during this period, why not inviting all members of the donors' household to join the study?

The movement and other restrictions in place also prevented this from happening, and the blood donation service would not have been able to implement this sampling. This explanation can be found on page 6 lines 70-76.

Our priority for when restrictions can be overcome would be the gold-standard of a probability-based population sample.

5. The gender disequilibrium between blood transfusion samples and National Census is very concerning as well as the fact that very small regions (census speaking) are largely represented in the transfusion samples.

It is true that blood donors are predominantly male in Kenya. However, we assayed ~2000 female donor samples and stratified all analyses by sex to ensure that any potential confounding was appropriately adjusted for. This can be found on page 6 lines 64-66

We interpret the regional distribution differently. The samples available are well distributed across regions for example, Rift Valley, Nyanza and Coast (including Mombasa), Central, Western and Nairobi (see table 1A).

6. Authors should further comment and consider further adjustment to reassure the readers. For example, did the authors post-stratify the modelled results accounting for the age and sex distribution to generate population-representative seroprevalence estimates?

In the results and discussion section on page 4 lines 33-36, we describe the adjustments made to the data including accounting for the gender disequilibrium. "We used Bayesian Multi-level Regression with Post-stratification (MRP) to adjust for test sensitivity (93%) and specificity (99%)⁶, smooth trends over time, and account for the differences in age, sex and residence characteristics of the test sample and the Kenyan population⁷."

7. More information is now available on the waning function of antibodies, adjusting some of the results would be outstanding.

The waning function of antibodies in our study or similar setting is still unknown and waning adjustments in other settings have proven controversial (page 5 line 58, reference 12). Rather than adjust our results for this through modelling, we have opted to present them as they are. We have also used our data to explore the application of mixture modelling approaches to account for this challenge elsewhere (page 5 lines 59-60, reference 13). Finally, we acknowledge our seroprevalence results may be an underestimate of cumulative incidence in Kenya (page 5 lines 60-61).

8. References are needed in the first paragraph, not clear where the data and comments are coming from.

We have now provided additional references on page 1 lines 3 and 8

9. More information on moving restriction policy in Kenya during the study period is needed

This has now been addressed on page 6 line 71 through reference 8.

10. WHO recommends monitoring changes of seroprevalence over time to plan an adequate public health response, what was the response in Kenya (physical distancing and confinement measures)?

Physical distancing and restrictions on activity have been used in Kenya, and these have been done taking into account the rates of severe disease and death as well as seroprevalence. More details on the measures are as described in reference 8 on pages 5 line 51 and 6 line 71.

11. Even though it is mentioned in the supplement I suggest to report the age of eligibility for donors (16-65)

This information has now been provided on page 3 line 14

12. Not clear whether the samples were consecutive cases.

This information can be found on page 8 line 121

13. Can the authors provide a reference on the assay performances they reported?

This is shown on page 4 line 27 through reference 3.

14. Here readers should understand when the second wave started in Kenya, and should have a better understanding of the dynamic of cases during the periods

We agree this is useful information. It is beyond the subject of the manuscript under review which is about the first epidemic wave. We will be able to share data on the second and other waves as we continue our surveillance in blood donors and other populations, but these data are not yet available. We touch on this on page 6 lines 74-75.

15. What is planned to assess the seroprevalence among young adults and elderly?

There are now covered in other surveys by us and other research groups.

16. Is it correct that there was no IRB approval needed?

Page 9 lines 146-149 shows IRB approval and written informed consent were obtained

17. Did the authors use a confirmatory test (recombinant immunofluorescence assay) potentially indeterminate individuals?

We did not conduct any further confirmatory testing and there were no potentially indeterminate individuals with our assay.

18. Important to stress that model did not account for random effect for household

We did not sample by household and did not conduct any random effects regression analyses. As described in the methods on Page 9 lines 132-140, we used Bayesian Multi-level Regression with Post-Stratification to account for differences in the age and sex distribution of blood donors and differences in the numbers of samples collected over time by region/location.

19. Did the authors run iterations and assessed convergence in their models?

We simulated 10,000 iterations of 3 chains with a burn in of 1,000 iterations. Convergence was assessed through the R-hat statistic and by visual inspection of the chains. This is now mentioned in the supplementary material on page 8.

20. Why didn't the authors calculate and provide the relative risk (RR) of being seropositive?

Our main objective for this paper was to describe the dynamics of SARS-CoV-2 seroprevalence among blood donors in Kenya during the course of the first epidemic wave.

Did the authors consider using their age-specific seroprevalences estimate the infection fatality risk?

As indicated on page 5 line 52, this is covered in another paper i.e., reference 8.

21. Figure 1:

Estimate should include CI (this would help interpret the distance between raw and modeled estimations)

Confidence intervals are now included in this figure as described in the legend on page 11 line 165-165

22. Authors should comment on the major distance between raw and modeled estimates (clearly illustrated in Figure 1).

As the reviewer seems to be suggesting, the discrepancy between modelled and observed estimates is mainly a function of sample size: the discrepancy is greatest for periods where few samples were available. This is illustrated more clearly now through the inclusion of confidence intervals in Figure 1.

23. Suppl Table 2:
the voluntary and family subtype must be reported in the methods

This was reported on page 8 lines 117-120

24. Table 1a:
Suggest to add % to the All samples column

This has been done